# RNA Editing: A New Therapeutic Target in Amyotrophic Lateral Sclerosis and Other Neurological Diseases

**DOI:** 10.3390/ijms222010958

**Published:** 2021-10-11

**Authors:** Takashi Hosaka, Hiroshi Tsuji, Shin Kwak

**Affiliations:** 1Department of Neurology, Division of Clinical Medicine, Faculty of Medicine, University of Tsukuba, Tsukuba 305-8575, Ibaraki, Japan; hosaka-ehi@umin.ac.jp (T.H.); htsuji@md.tsukuba.ac.jp (H.T.); 2Department of Internal Medicine, Tsukuba University Hospital Kensei Area Medical Education Center, Chikusei 308-0813, Ibaraki, Japan; 3Department of Internal Medicine, Ibaraki Western Medical Center, Chikusei 308-0813, Ibaraki, Japan; 4Department of Neurology, Tokyo Medical University, Shinjuku-ku, Tokyo 160-0023, Japan

**Keywords:** amyotrophic lateral sclerosis (ALS), excitotoxicity, RNA editing, α-amino-3-hydroxy-5-methyl-4-isoxazole propionic acid (AMPA) receptors, adenosine deaminase acting on RNA 2 (ADAR2)

## Abstract

The conversion of adenosine to inosine in RNA editing (A-to-I RNA editing) is recognized as a critical post-transcriptional modification of RNA by adenosine deaminases acting on RNAs (ADARs). A-to-I RNA editing occurs predominantly in mammalian and human central nervous systems and can alter the function of translated proteins, including neurotransmitter receptors and ion channels; therefore, the role of dysregulated RNA editing in the pathogenesis of neurological diseases has been speculated. Specifically, the failure of A-to-I RNA editing at the glutamine/arginine (Q/R) site of the GluA2 subunit causes excessive permeability of α-amino-3-hydroxy-5-methyl-4-isoxazole propionic acid (AMPA) receptors to Ca^2+^, inducing fatal status epilepticus and the neurodegeneration of motor neurons in mice. Therefore, an RNA editing deficiency at the Q/R site in GluA2 due to the downregulation of ADAR2 in the motor neurons of sporadic amyotrophic lateral sclerosis (ALS) patients suggests that Ca^2+^-permeable AMPA receptors and the dysregulation of RNA editing are suitable therapeutic targets for ALS. Gene therapy has recently emerged as a new therapeutic opportunity for many heretofore incurable diseases, and RNA editing dysregulation can be a target for gene therapy; therefore, we reviewed neurological diseases associated with dysregulated RNA editing and a new therapeutic approach targeting dysregulated RNA editing, especially one that is effective in ALS.

## 1. Introduction

Amyotrophic lateral sclerosis (ALS), also known as Lou Gehrig’s disease in the USA, is an adult-onset fatal motor neuron disease characterized by the degeneration of the upper (cortical) and lower (spinal and brain stem) motor neurons, resulting in progressive muscular weakness and ultimately death due to respiratory failure [1]. Approximately 90% of ALS cases are sporadic, with ALS not being present among their blood relatives, and approximately only 10% of the cases are familial. Recent surveillance has identified more than 30 ALS-linked genes, including *superoxide dismutase 1* (*SOD-1*), *transactive response DNA*/*RNA binding protein of 43 kDa* (*TDP-43*), *fused in sarcoma (FUS),* and *chromosome 9 open reading frame 72* (*C9ORF72*) [2]. The incidence of ALS is 2 to 3 people per a 100,000-person population and is increasing in developed countries due to an increase in life expectancy. Although various plausible mechanisms such as excitotoxicity due to the dysregulation of glutamatergic signaling, axon transport impairment, neuroinflammation, protein aggregation, or oxidative stress have been proposed as the etiology of ALS, no one has been successful in elucidating the mechanisms underlying the molecular and morphological changes in the degenerating motor neurons of ALS patients [3,4]. Although two drugs, riluzole and edaravone, have been approved for the treatment of ALS, neither can significantly extend the lives of patients [2,5].

Neurons in the central nervous system (CNS) use glutamate, an excitatory neurotransmitter. Depolarization of pre-synaptic neurons (upper motor neurons) leads to the vesicular release of glutamate into the synaptic cleft; after secretion from the axon terminal of the upper motor neurons, glutamate activates the post-synaptic neurons (lower motor neurons) through specific membrane glutamate receptors [6]. Glutamate receptors (GluRs) are classified into two major receptor families: ionotropic GluRs and metabotropic GluRs (mGluRs) [7]. Ionotropic GluRs are ligand-gated ion channels that are composed of four subunits that mediate an immediate influx of extracellular Na^+^ and/or Ca^2+^, regulating membrane depolarization and diverse signal transduction events [8]. Alpha-amino-3-hydroxy-5-methyl-4-isoxazole propionic acid (AMPA) receptors regulate the membrane potential and mediate the vast majority of fast excitatory neurotransmission, whereas N-methyl-D-aspartate (NMDA) receptors have a voltage-dependent ion channel that mediates excitatory neurotransmission by passing large amounts of Ca^2+^, Na^+^, and K^+^ during depolarization [8,9,10]. In contrast, mGluRs are G protein-coupled receptors and are subclassified into three groups based on sequence homology. Groups I, II, and III include mGluR1 and mGluR5; mGluR2 and mGluR3; and mGluR4, mGluR6, mGluR7, and mGluR8, respectively. In pre-synaptic neurons, Group I mGluRs promote glutamate release, whereas Group II and III mGluRs inhibit glutamate release. In post-synaptic neurons, mGluRs act relatively slowly compared to ionotropic GluRs, and Group I mGluRs increase intracellular Ca^2+^ influx via G protein signaling cascades [7,11].

Excitatory signals are terminated by the uptake of glutamate released into the surrounding neurons and/or astrocytes through the glutamate transporters and excitatory amino acid transporters (EAATs) [3,6]. Five members, EAAT1 to EAAT5, are known in the EAAT family: EAAT1 or GLAST is the major glutamate transporter and is primarily expressed in the Bergmann glial cells of the cerebellar cortex; EAAT2, or GLT-1, is responsible for more than 90% of glutamate uptake and is expressed primarily in the astrocytes of the cerebral cortex and hippocampus; EAAT3, or excitatory amino acid carrier 1 (EAAC1), is primarily expressed in the pre-synaptic neurons of the cerebral cortex and basal ganglia, while EAAT4 and EAAT5 are low-capacity glutamate transporters expressed in the post-synaptic dendrites of the Purkinje cells of the cerebellar cortex and the amacrine cells of the retina, respectively [6]. These transporters rapidly re-uptake the released glutamate, thereby maintaining its concentration in the synaptic cleft at sufficiently low levels.

Exaggerated activation of the glutamatergic system leads to the hyperexcitation of neurons and ultimately excitotoxicity. Elevated glutamate levels in the post-mortem tissue and cerebrospinal fluid of ALS patients [12,13,14,15] and the loss of high-affinity glutamate uptake [16] makes excitotoxicity an attractive therapeutic target for ALS. Although an earlier trial using branched-chain amino acids, a modifier of glutamate metabolism, turned out to be ineffective [17], riluzole, an inhibitor of glutamate release, was approved for use as an ALS drug. Riluzole improved one-year survival rates and progressive muscle weakness [18,19,20], especially in the late stages of ALS [21]. However, the effects of riluzole are limited. The one-year survival was only 9%, and the median survival was prolonged by only 2–3 months [22]. In addition, studies using transcranial magnetic stimulation (TMS) or threshold tracking nerve conduction studies (TTNCSs) have shown cortical and spinal motor neuron hyperexcitability in sporadic and familial ALS patients, and cortical hyperexcitability due to increased glutamatergic drive has been hypothesized [23,24,25,26]. Ezogabine, an activator of Kv7 potassium channels, reduced neuronal excitability and improved the in vitro survival of differentiated motor neurons from induced pluripotent stem cells (iPSCs) derived from ALS patients and decreased cortical and spinal motor neuron excitability, as evaluated by TMS and TTNCSs in ALS patients [27,28]. However, it is unclear whether a reduction in cortical hyperexcitability prolongs survival in patients with ALS. Other potential ALS drugs targeting neuronal hyperactivity or excitotoxicity, including memantine, a non-competitive antagonist of NMDA receptors, lamotrigine, which inhibits glutamate release and inactivates voltage-gated calcium channels, and talampanel, a non-competitive antagonist of AMPA receptors, were found to be unsuccessful in clinical trials involving patients with ALS [29,30]. Therefore, different therapeutic approaches are needed to effectively prevent neuronal death in ALS.

Mechanistically, exaggerated Ca^2+^ influx through glutamate receptors plays a pivotal role in excitotoxicity [3]. Ca^2+^ plays an important role in physiological neuronal functions, including those related to synaptic plasticity, presynaptic transmitter release, and postsynaptic responses as a second messenger; however, exaggerated Ca^2+^ influx results in brain damage (Figure 1) [7,8]. Exaggerated activation of NMDA receptors is involved in rapid cell death in diseases such as epilepsy and encephalitis [31,32]. Though most AMPA receptors do not mediate large Ca^2+^ influx, a small proportion are Ca^2+^-permeable, and these AMPA receptors mediate the slow death of motor neurons, which is reminiscent of the death of motor neurons in ALS [4,33].

AMPA receptors are comprised of homo- or hetero-tetramers of GluA1, GluA2, GluA3, and GluA4, of which only GluA2 is subjected to adenosine-to-inosine (A-to-I) RNA editing at the glutamine/arginine (Q/R) site [34,35]. GluA2 is the determinant of the Ca^2+^ permeability of AMPA receptors, and all of the GluA2 expressed in the neurons is edited at the Q/R site by adenosine deaminase acting on RNA 2 (ADAR2) [36]. AMPA receptors are Ca^2+^-impermeable when they contain edited GluA2 in their subunit assembly, whereas AMPA receptors devoid of GluA2 or containing unedited GluA2 are Ca^2+^-permeable [37]. Additionally, editing at the Q/R site in GluA2 reduces the AMPA receptor tetramerization and trafficking of GluA2 to the synaptic membrane, suggesting that Q/R site-unedited GluA2 is more prone to being integrated into functional AMPA receptors than edited GluA2 is, enhancing Ca^2+^ influx through the AMPA receptors [38]. Moreover, auxiliary subunits of AMPA receptors, such as transmembrane AMPA receptor regulatory protein (TARP) and cornichon family AMPA receptor auxiliary protein 2 (CNIH2), have a regulatory role in Ca^2+^ influx through AMPA receptors by influencing gate dilation [39,40].

In this report, we summarized current knowledge regarding the failure of A-to-I RNA editing in neurological diseases. Moreover, as some recent excellent reviews have described the association of RNA editing with neurodegenerative diseases or current and promising gene therapies for neurodegeneration in general [41,42,43,44], we focused on ALS-related excitotoxicity due to the dysfunction of glutamatergic signaling resulting from dysregulated RNA editing at the GluA2 Q/R site, which are promising future targets in ALS therapy.

## 2. RNA Editing and Neurological Diseases

A-to-I RNA editing, or site-specific and post-transcriptional modification of RNA by ADARs, occurs in various classes of RNA, including both protein-coding and non-coding mRNAs, microRNAs (miRNAs), and circular RNAs (circRNAs) [45]. Among the millions of A-to-I sites, the vast majority reside in the non-coding regions of RNA, including in the 5′- and 3′-untranslated regions (UTRs) and the *Alu* repetitive elements or short interspersed nuclear element (SINE) sequences [46], whereas less than 1 % reside in the coding regions in mammals and humans [45,47,48]. A-to-I sites are the most abundant in the RNAs expressed in the CNS, and several mRNAs that code neuronal ion channels and receptors have A-to-I sites in the coding regions. The editing efficiency varies widely from less than 1% to 100% among different RNAs; at the same, the A-to-I site in the same RNA differs among the various tissues, developmental stages, environmental conditions, and cell types [49]. A-to-I RNA editing can influence the efficacy of RNA splicing, translation, localization, and the stability of RNA and biogenesis of non-coding RNAs, including miRNA and circRNA [50,51,52]. Moreover, RNA editing within protein-coding sequences can form mRNA that carries a codon that is different from the original DNA, thereby altering protein structure and function [53,54]. There are three ADAR proteins in humans: ADAR1, ADAR2, and ADAR3, which are are encoded by *ADAR*, *ADARB1*, and *ADARB2*, respectively. ADAR1 and ADAR2 have editing activity in vitro and in vivo, whereas ADAR3 has no natural substrate RNA and has a possible role as a negative regulator of RNA editing by sequestering the editing substrate of ADAR1 and ADAR2 [45,55,56]. ADAR1 is widely expressed across tissues, and ADAR1p110, a 110 kDa isoform protein, localizes primarily in the nucleus, while ADAR1p150, a 150 kDa isoform protein, localizes in the cytoplasm [57]. ADAR2 is also widely expressed, but its expression level in the CNS is higher than that of ADAR1; ADAR3 is expressed exclusively in the brain, in the glial cells in particular [47,56,58].

ADARs have distinct roles in the development and expression of normal phenotypes in mammals and humans. ADAR1 is crucial for developing non-nervous tissues, including the liver and spleen, and mice lacking ADAR1 are lethal to embryos due to widespread apoptosis [59]. Mutations in *ADAR* are principally associated with dyschromatosis symmetrica hereditaria and Aicardi–Goutieres syndrome [60,61]. In contrast, mice lacking ADAR2 develop normally but die by three weeks of age from recurrent epileptic seizures [62]. Infants carrying bi-allelic variants of *ADARB1*, the gene encoding ADAR2, develop either epileptic encephalopathy or microcephaly associated with intellectual disability and seizures [50,63,64]. In addition to the roles in developing tissues, the conditional targeting of *ADARB1* in mouse motor neurons led to their slow death [65], indicating that A-to-I RNA editing plays an important role in proper neuronal functions in mature CNS [66]. Therefore, the dysregulation of A-to-I RNA editing has been suggested to be associated with several adult- or adolescent-onset neurological diseases, including ALS, epilepsy, and schizophrenia [48,67,68,69,70] (Table 1).

Among the neurological diseases associated with defects in RNA editing, the pathogenic roles of A-to-I RNA editing dysfunction in the GluA2 Q/R site due to the downregulation of ADAR2 have been extensively demonstrated in sporadic ALS, as described in the next section [36,71]. A comprehensive study on RNA editing in the tissues of post-mortem Alzheimer’s disease (AD) patients reported a reduction in RNA editing activity at various A-to-I sites in the hippocampus, temporal lobe, and frontal lobe [72], and neuronal death in AD has been suggested to be associated with disturbance of RNA editing at the GluA2 Q/R site [73,74]. A comprehensive analysis of RNA editing alterations in glioblastoma or Grade IV glial cell tumors reported significant changes in editing efficiencies at a large number of RNA editing sites. Reduced A-to-I RNA editing at the Q/R site of GluA2 mRNA has a role in the proliferation and migration of glioblastoma cells via the activation of the Akt pathway [75,76,77]. The reduction in GluA2 Q/R site RNA editing in the glioblastoma cells results from the overexpression of ADAR1 and ADAR3, which may inhibit the homodimerization of ADAR2 rather than from a reduction in ADAR2 expression level [77]. Malignant glioma is also associated with a reduction in RNA editing at various A-to-I sites, including the isoleucine/methionine (I/M) site in gamma-aminobutyric acid type A receptor subunit alpha 3 (GABRA3) [78], five editing sites in cell division cycle 14 B (CDC14B) [79], the +9 site of miR-336, and the +6 site of miR-589-3p [80,81].

Another neurological disease associated with deficient RNA editing is brain ischemia, in which an aberrant Ca^2+^ influx through Q/R site-unedited GluA2 containing Ca^2+^-permeable AMPA receptors has been suspected to play a crucial role in the neuronal death of CA1 pyramidal neurons in mice [82,83]. RNA editing at the arginine/glycine (R/G) site in GluA2 mRNA is reduced in the core and penumbra of acute spinal cord injury [84]. Moreover, the RNA editing at site D in 5-hydroxytryptamine receptor 2C (5-HT2_C_), a subtype of serotonin receptors, and the isoleucine/valine (I/V) site in Kv1.1 have been reported in injured rat spinal cords [85]. Although the factors relevant to abnormal RNA editing in spinal cord injury are unclear, these changes in RNA may influence postsynaptic excitatory responses to glutamate, culminating in the modulation of cell death progression [84].

Since severe epileptic seizures induce brain damage, the role of aberrant RNA editing in patients with epilepsy has been investigated. Epilepsy is characterized by abnormal neuronal hyperexcitability and affects over 70 million people worldwide [86]. Although mice expressing Q/R site-unedited GluA2 exhibited recurrent epileptic seizures and early postnatal death [62], RNA editing at this site was not reduced in the brains of postmortem patients [62,87]. In contrast, increased editing efficiency has been recognized at the R/G site in GluA2, the Q/R site in glutamate ionotropic receptor kainite type subunit 1 (GluK1), and the Q/R site in GluK2 in surgically excised brain tissues of patients with epilepsy [87,88,89]. These changes enhance the response to glutamate and modulate neuronal excitability in the mouse brain [62,90,91].

Changes in monoamines such as serotonin and dopamine have been implicated in common psychiatric disorders, including depression and schizophrenia [92]. Although the influence of changes in editing efficiencies at each of the five editing sites in 5-HT2_C_ were inconsistent among previous reports [93,94,95,96,97,98,99], the proportion of site A-edited 5-HT2_C_ was higher, especially in the postmortem brains of suicide victims than in non-suicidal patients with both depression and schizophrenia [96,99]. Interestingly, editing efficiencies at sites B, C, and E in phosphodiesterase 8A (PDE8A), a key modulator of signal transduction downstream of 5-HT2_C_, were decreased in the brain and blood of patients with depression [100,101]. A comprehensive study on the alteration of RNA editing has demonstrated a higher overall RNA editing level and significant changes in editing efficiencies at many editing sites in the brains of patients with schizophrenia [70].

Comprehensive studies on alterations in RNA editing have demonstrated changes in editing efficiencies at many editing sites in the brains of patients with autism spectrum disorder, a common neurodevelopmental disorder characterized by social communication deficits and repetitive sensorimotor behaviors [102]; however, no specific editing site relevant to the pathogenesis could be identified [69,103]. In addition, studies in mice and rats have shown negative effects of alcohol and/or cocaine abuse/abstinence on the RNA editing efficiency of ADAR2 [104,105].

## 3. Excitotoxicity in ALS Due to Excessive Ca^2+^ Influx through Unedited GluA2-Conaining AMPA Receptors

As described in the introduction section, the role of excitotoxicity, mediated by Ca^2+^-permeable AMPA receptors, has been proposed in ALS pathogenesis [4,106,107] (Figure 2). AMPA receptors are Ca^2+^-permeable when they either lack GluA2 or contain Q/R site-unedited GluA2 in their subunit assembly.

Studies on mutant mice have demonstrated that the expression of unedited GluA2 induces fatal epilepsy in mice [62] and that the expression level of unedited GluA2 correlates with the level of Ca^2+^ influx and the severity of neurological dysfunctions [62,108,109,110]. Conversely, a study on GluA2 knockout mice indicated that expression of GluA2-lacking Ca^2+^-permeable AMPA receptors per se does not cause the death of motor neurons [111]. Notably, in conditional ADAR2 knockout (ADAR2*^flox^*^/*flox*^/VAChT.Cre or AR2) mice, the motor neurons undergo slow progressive death when the process of RNA editing at the Q/R site of GluA2 is ablated selectively in these cells [65]; this suggests that the expression of Q/R site-unedited GluA2 rather than a decreased expression of GluA2 plays a central role in neuronal death, particularly motor neuron death, mediated by Ca^2+^-permeable AMPA receptors. In the spinal motor neurons of patients with sporadic ALS, the Q/R site-unedited GluA2 is expressed due to downregulation of ADAR2 [36,71], suggesting the pathogenic role of unedited GluA2 expression in ALS. However, there have been several contradictory reports. The lack of expression of unedited GluA2 in the whole spinal cord tissue of sporadic ALS patients [112] may be due to the masking of the small proportion of unedited GluA2 by the abundantly expressed edited GluA2; moreover, the proportions of unedited GluA2 differ markedly (from 0% to 100%) among individual motor neurons in ALS patients when analyzed at the single-cell level [36,71]. The lack of unedited GluA2 in iPSC-derived motor neurons from sporadic ALS or familial ALS patients carrying *FUS^H517D^* mutations [113,114] may be due to either or both of the following reasons: the considerably shorter life time of iPSC-derived differentiated motor neurons in culture than that of the degenerating motor neurons in ALS patients or the large difference in the cellular environment between the in vitro culture condition and the in vivo microenvironmental conditions surrounding the degenerating motor neurons in patients.

Moreover, the pathogenic role of Q/R site-unedited GluA2 in ALS is demonstrated by its role in TDP-43 pathology, the pathological hallmark of ALS. TDP-43 pathology is observed exclusively in motor neurons lacking ADAR2 in patients with sporadic ALS [115]. Studies on conditional ADAR2 knockout AR2 mice have demonstrated that TDP-43 mislocalizes from the nucleus to the cytoplasm in a manner that is dependent on increased Ca^2+^ influx due to the expression of unedited GluA2 in the motor neurons lacking ADAR2 [116]. Mechanistically, an increase in intracellular Ca^2+^ levels activates calpain, which cleaves TDP-43 into aggregation-prone fragments; the resultant aggregates progressively enlarge with continued calpain activation caused by elevated intracellular Ca^2+^ resulting from the expression of Ca^2+^-permeable AMPA receptors [116,117]. These results suggest that progressive ADAR2 downregulation determines the initiation and progression of sporadic ALS (ADAR2-GluA2 hypothesis, [118]) (Figure 3).

In addition to sporadic ALS, some ALS-linked genes are involved in excitotoxicity due to the dysregulation of glutamatergic signaling or to A-to-I RNA editing.

FUS is involved in RNA processing, and it has been demonstrated that excitotoxicity affects FUS translocation from the nucleus to the cytoplasm and increases FUS-mediated dendritic GluA2 mRNA expression [119]. Additionally, glutamate release from brain synaptosomes and glutamate uptake were increased in transgenic mice with mutant *FUS* lacking a nuclear localization signal motif [120]. Regarding RNA editing, ADAR2 downregulation with a concomitant expression of Q/R site-unedited GluA2 mRNA was reported in the motor neurons of ALS patients carrying the *FUS*^P525L^ mutation [121] but not in motor neurons differentiated from iPSCs derived from ALS patients carrying *FUS^H517D^* mutation [113].

Furthermore, reduced ADAR2 activity due to the loss of nuclear ADAR2 rather than a decrease in ADAR2 expression was reported the cause of a widespread reduction in RNA editing with ADAR2 cytoplasmic mislocalization in the motor neurons and differentiated motor neurons, which were generated from iPSCs derived from ALS patients carrying *C9ORF72* with enhanced hexanucleotide repeat expansion [122]. Moreover, it has been demonstrated that dipeptide repeat proteins derived from the hexanucleotide repeat expansion of *C9ORF72* bind to ADAR1 and ADAR2, thereby inhibiting their RNA editing activity in vitro [123]. However, one study reported that all GluA2 was edited at the Q/R site in the motor neurons differentiated from iPSCs derived from ALS patients carrying *C9ORF72* with enhanced hexanucleotide repeat expansion [124].

Although TDP-43 mislocalizes from the nucleus to the cytoplasm in the primary neuronal culture exposed to excitotoxicity [119] and may have a modulatory role in ADAR1-mediated A-to-I RNA editing in the cell lines [125], no studies have demonstrated aberrant RNA editing at the Q/R site in GluA2 in familial ALS carrying *TARDBP* mutations. In addition, possible roles of excitotoxicity in ALS-linked *SOD-1* mutations have been suggested in *SOD-1* transgenic mice [126,127], but all of the GluA2 mRNA expressed in the motor neurons of mutated human *SOD-1* transgenic rats were edited at the Q/R site [128].

The close association of reduced ADAR2 activity with both motor neuron death and TDP-43 pathology in sporadic and some forms of familial ALS cases lends further support to the ADAR2-GluA2 hypothesis in ALS pathogenesis.

TDP-43 pathology was also observed in the motor neurons of *TARDBP*-linked ALS patients. However, a reduction in ADAR2 activity does not appear to be involved in the mechanism underlying the death of motor neurons or TDP-43 pathology. In the motor neurons, mutant TDP-43 is more vulnerable to calpain-dependent cleavage than wild-type TDP-43 is and is readily cleaved by activated calpain; GluA2-lacking Ca^2+^-permeable AMPA receptors are expressed more abundantly in the motor neurons than in other neuronal classes, such as cortical neurons [116]. Thus, Ca^2+^-permeable AMPA receptors are involved in the pathogenic mechanism underlying TDP-43 pathology in *TARDBP*-linked ALS in a way different from that in other ALS types.

An increase in the proportion of GluA2-lacking Ca^2+^-permeable AMPA receptors is associated with hyperactivity in ALS motor neurons (Figure 2); this hyperactivity was associated with an increase in the expression levels of GluA1 mRNA in the spinal motor neurons of *C9ORF72* ALS patients [114] and in *FUS* knockdown mice [129,130], an increase in the expression levels of GluA1 mRNA and GluA3 mRNA in the iPSC-derived motor neurons of *C9ORF72* ALS patients [124,131], a decrease in GluA2 mRNA in *FUS* knockdown mice or differentiated motor neurons derived from the embryonic stem cells of *FUS^P525L^* knock-in mice [129,130], and an increase in GluA4 mRNA expression in the iPSC-derived motor neurons from familial ALS patients carrying the *FUS^R521C^* and *TARDBP^M337V^* mutations [132]. However, as GluA2 knockout mice do not exhibit any neuronal loss [111], it is unclear whether a relative reduction of GluA2, among the other AMPA receptor subunits, is mechanistically associated with motor neuron death in ALS.

Therefore, these results indicate that excessive Ca^2+^ influx through Ca^2+^-permeable AMPA receptors containing unedited GluA2 is critically involved in the pathogenesis of both sporadic and familial ALS.

## 4. Promising Therapy Targeting RNA Editing Dysregulation in Neurological Diseases

As a reduction in RNA editing activity is involved in the pathogenesis of neurological diseases (Table 1), the normalization of RNA editing activity in the motor neurons is a therapeutic strategy for not only sporadic ALS and some familial ALS but also various neurological diseases for which no cure can be achieved by current therapies (Figure 4).

Restoration of ADAR2 activity is a promising fundamental therapy for sporadic ALS and some forms of familial ALS. Elevated ADAR1 expression promotes cancer growth and metastasis [133], whereas in *A**DARB**1* transgenic mice, elevated ADAR2 expression is associated with simple obesity due to chronic hyperphagia, but apparently normal motor, lung, and heart functions [134]. Indeed, the delivery of ADAR2 cDNA to the motor neurons of conditional ADAR2 knockout AR2 mice via adeno-associated virus serotype 9 (AAV-9) robustly prevented progressive motor dysfunction and motor neuron death and also improved TDP-43 mislocalization without adverse behavioral or pathologic effects [135]. An increasing number of clinical trials using gene therapy for human diseases, including neurological diseases, have been conducted, and some of them have been approved for clinical use [136,137]. Although gene therapy is associated with several problems, such as daunting costs and regulatory policies, AAV9-ADAR2 therapy is a promising fundamental treatment for sporadic ALS in the future.

RNA base editing using clustered regularly interspaced short palindromic repair (CRISPR)-Cas13 could also be a potential therapeutic approach for deficient RNA editing-associated neurological diseases, including ALS. Programmable A-to-I RNA editing using CRISPR-Cas13-fused hyperactive mutant ADAR2 has proven to be successful in the site-directed A-to-I editing of point mutations in human disease genes such as *methyl-CpF binding protein 2 (MECP2)* (311G>A) in Rett syndrome and *survival motor neuron (SMN1)* (305G>A) in spinal muscular atrophy in vitro. The development of more efficient and safer RNA base editors [138,139,140] as well as more efficient delivery systems targeting diseased cells—such as AAV vectors [138], will realize the use of programmable A-to-I RNA editing using CRISPR-Cas13-fused hyperactive ADAR2 mutant as a future therapeutic strategy for deficient RNA editing-associated neurological diseases, including ALS.

Since the death of ADAR2-deficient motor neurons in AR2 mice is specifically mediated by Ca^2+^-permeable AMPA receptors, AMPA receptor antagonists are promising candidates to treat ALS. Non-competitive AMPA receptors have been demonstrated to confer protection against motor neuron death caused by excitotoxicity and 1,2,3,4-tetrahydro-6-nitro-2,3-dioxo-benzo[f]quinoxaline-7-sulfonamide disodium salt (NBQX)—the selective AMPA/kainate receptor antagonist, as well as prolong the survival of transgenic *SOD1^G93A^* mutant mice [141,142]. Moreover, in other studies, perampanel, an FDA-approved non-competitive AMPA receptor antagonist to manage epilepsy, has prevented ALS-like progressive motor dysfunction in conditional *ADAR2* knockout mice and increased the cortical excitability threshold [143,144]. However, with the exception of an improvement in the manual muscle testing score, perampanel did not effectively inhibit disease progression in a recent phase 2 clinical trial on sporadic ALS patients [145,146]. The clinical use of AMPA receptor antagonists is limited by their frequent adverse effects due to the suppression of the physiological function of the CNS neurons; therefore, the development of more sophisticated drugs, such as receptor subunit-specific antagonists, are required in the future.

A-to-I RNA editing-based therapeutic strategies would effectively rescue the death of lower motor neurons in the majority of ALS patients, including patients with both sporadic and some with familial ALS that are associated with the dysregulation of A-to-I RNA editing. However, as the roles of dysregulation of RNA editing in the upper motor neuron death remain elusive, further studies are needed for the elucidation of the pathogenic mechanism underlying the death of the upper motor neurons in ALS and, more broadly, the death of the cortical neurons in frontotemporal lobar degeneration, which is closely associated with ALS. When such a new therapeutic strategy is realized in ALS, disease-specific biomarkers will also be increasingly required. Since biomarkers linked to disease-specific molecular changes can specify ALS, the demonstration of deficient ADAR2 activity non-invasively will help immensely in the evaluation of treatment efficacy. As the RNA editing efficiency at the ADAR2 sites in extracellular RNAs reflects intracellular ADAR2 activity in vitro [147], demonstrating a reduction in the editing levels of these RNAs in the cerebrospinal fluid would become a biomarker for ALS.

## 5. Conclusions

We reviewed the mechanistic roles of defective RNA editing in neurological diseases and the potential therapeutic strategies targeting RNA editing in these diseases, especially for ALS. As excitotoxicity mediated by excessive Ca^2+^ influx through Ca^2+^-permeable AMPA receptors seems to be a plausible pathomechanism in ALS, the development of therapy based on this underlying pathomechanism would change the currently incurable ALS to a treatable disease. Moreover, we will be able to evaluate therapeutic response noninvasively by measuring alterations in editing efficiency in the RNAs in body fluids.

## Figures and Tables

**Figure 1 ijms-22-10958-f001:**
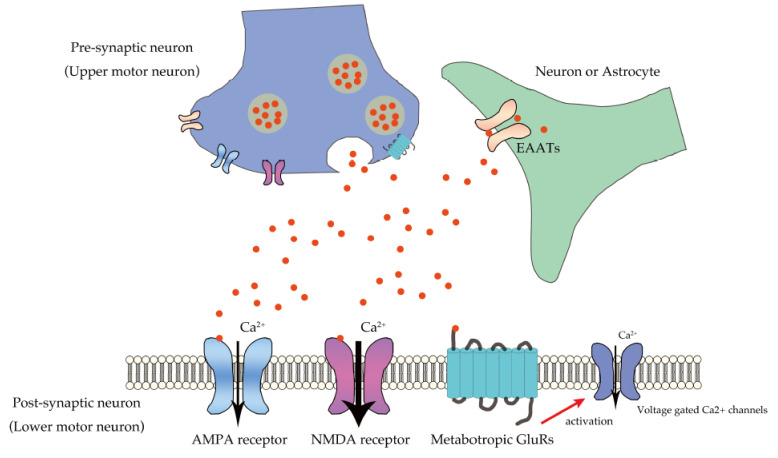
Ca^2+^ influx in lower motor neurons through glutamatergic neurotransmission. Glutamate secreted from pre-synaptic neuron (upper motor neuron) into the synaptic cleft binds to the glutamate receptors, including the AMPA receptor, NMDA receptor, and mGluRs expressed in the postsynaptic neurons (lower motor neurons). AMPA receptors mediate excitatory neurotransmission by regulating membrane potential, whereas NMDA receptors predominantly mediate large amounts of Ca^2+^ influx after depolarization, and mGluRs mediate Ca^2+^ influx via voltage-gated Ca^2+^ channels. Glutamate transporters, EAATs expressed in neurons or astrocytes, regulate glutamate concentration in the synaptic cleft.

**Figure 2 ijms-22-10958-f002:**
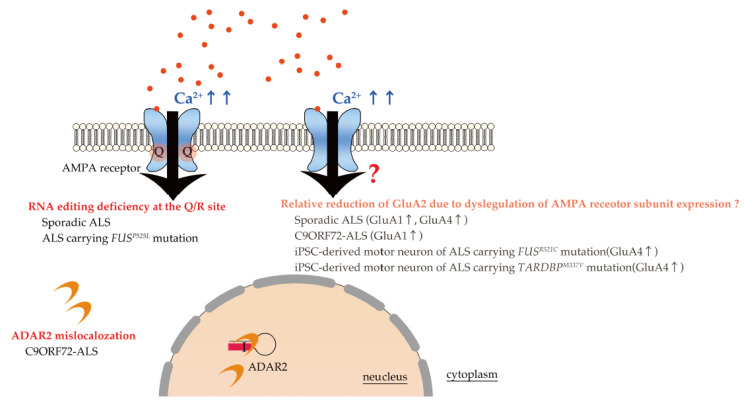
Excessive Ca^2+^ influx through Ca^2+^-permeable AMPA receptors in ALS. Ca^2+^ permeability of AMPA receptors (arrows) is determined by the presence or absence of Q/R site-edited GluA2 in the subunit assembly. RNA editing deficiency at the Q/R site in GluA2 due to downregulation of ADAR2 has been demonstrated in the motor neurons of patients with sporadic ALS and those carrying the *FUS*^P525L^ mutation. Furthermore, ADAR2 mislocalization with loss of RNA editing activity has been reported in C9ORF72-ALS patients. Alterations in the expression of various AMPA subunits has been reported in sporadic ALS, C9ORF72 patients, and familial ALS patients carrying the *FUS*^R521C^ and *TARDBP^M337V^* mutations; however, loss of GluA2 does not induce neuronal death in GluA2 knockout mice. Therefore, a relative reduction of GluA2 per se may not be pathological in causing motor neuron death in ALS. Then, the question mark near the arrow was added.

**Figure 3 ijms-22-10958-f003:**
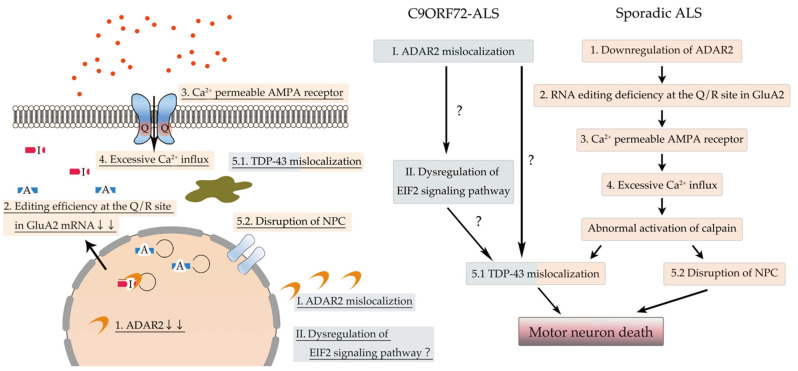
ADAR2-GluA2 hypothesis: specific molecular abnormalities beginning with a reduction in ADAR2 activity due to downregulation of ADAR2 in sporadic ALS or ADAR2 mislocalization in C9ORF72 ALS. The numbers and arrows indicate the order of molecular cascade based on the hypothesis. Uncertain parts were indicated by question marks.

**Figure 4 ijms-22-10958-f004:**
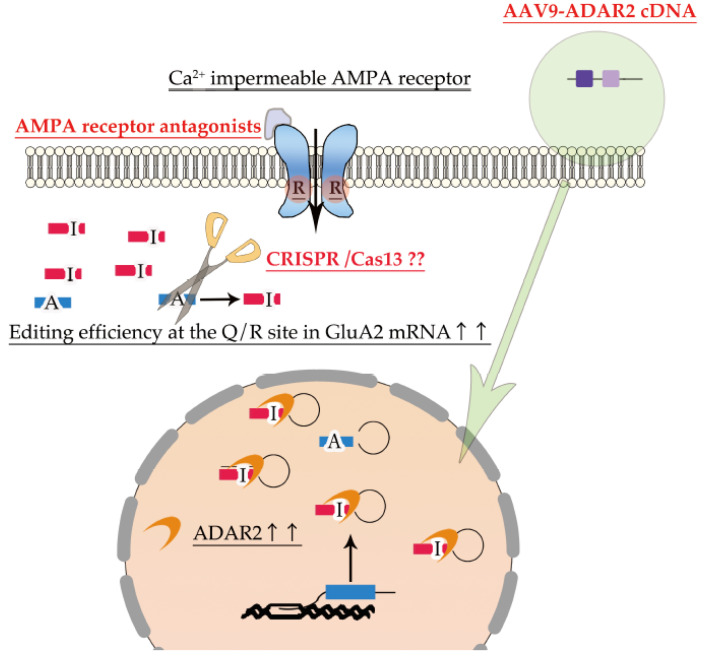
Promising therapy targeting dysregulation of ADAR2 RNA editing. AMPA receptor antagonists are potential therapeutic drugs for normalizing excessive Ca^2+^ influx through Ca^2+^-permeable AMPA receptors. Delivery of ADAR2 cDNA to motor neurons using the adeno-associated virus type 9 (AAV-9) vector is a therapeutic approach to improve the expression of ADAR2 in the degenerating neurons (green arrow). Furthermore, RNA base editing of unedited GluA2 using CRISPR-Cas13 could also be a potential therapeutic approach to increase the proportion of edited GluA2. As RNA base editing therapy have not been established, the question marks were added.

**Table 1 ijms-22-10958-t001:** CNS diseases linked to RNA editing.

Disease	Editing Site	Editing Efficiency	ADAR Responsible	Relation to Disease Pathogenesis	Pathogenetic Alteration	Reference
ALS	Q/R site in GluA2	Decrease	ADAR2	Exaggerated Ca^2+^ influx	Lower motor neurons (neuronal death, TDP-43 mislocalization)	[36,71]
AD	Q/R site in GluA2	Decrease	ADAR2	Not described	Not specified	[73,74]
Glioblastoma	Q/R site in GluA2	Decrease	ADAR2	Exaggerated Ca^2+^ influx Activation of Akt pathway	Malignant conversion (promotion of proliferation, migration, and invasion)	[75,76,77]
	I/M site in GABRA3	Decrease	ADAR1 ADAR2	Reduced inhibition of invasion and migration		[78]
	+9 site of miR-336	Decrease	ADAR2	Promotion of proliferation, migration, and invasion		[80]
	+6 site of miR-589-3p	Decrease	ADAR2	Promotion of proliferation, migration, and invasion		[81]
Ischemia	Q/R site in GluA2	Decrease	ADAR2	Exaggerated Ca^2+^ influx	Not specified	[82,83]
Spinal cord injury	R/G site in GluA2	Decrease	ADAR1 ADAR2	Reduction of postsynaptic excitatory response to glutamate	Not specified	[84]
	Site D in 5-HT2c I/V site in Kv1.1	Decrease	ADAR2	Not described		[85]
Epilepsy	R/G site in GluA2	Increase	ADAR1 ADAR2	Modulation of neuronal excitability	Not specified	[87,89]
	Q/R site in GRIK1 Q/R site in GRIK2	Increase	ADAR1 ADAR2	Exaggerated Ca^2+^ influx		[88]
Depression	Sites B, C, and E in PDE8A	Decrease	Unknown	Not described	Not specified	[100,101]

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
