# Peer review of "RNA Editing: A New Therapeutic Target in Amyotrophic Lateral Sclerosis and Other Neurological Diseases"

_ijms, 2021, doi:10.3390/ijms222010958_

Round 1
Reviewer 1 Report
Thank you for the opportunity to review this very interesting manuscript. This manuscript reviews the evidence to support the potential of A-to-I RNA editing based therapeutics to treat ALS, including covering the implication of glutamate excitotoxicity and RNA editing in neurological diseases and how this suggests this could be a promising therapeutic avenue. I have a few comments I think should also be considered in this space:
SOD1 has roles in glutamate excitotoxicity, and many other ALS genes such as TARDBP and FUS have roles in RNA processing. Could you include some discussion around these genes (and their encoded proteins) and their roles in the interaction of these two pathways? Could you also discuss any established or potential links (eg. are there any genomic features of these genes that flag them as being susceptible?) between ALS genes involved in RNA processing and A-to-I RNA editing?
Can you comment on whether you think A-to-I RNA editing based therapeutics will be beneficial to all ALS patients or only a subset? If only a subset, how would you determine which patients would or would not benefit, and can this be effectively translated into a routine clinical test for patients?
Could you provide a comment around whether you think this sort of therapeutic approach would be applicable to the closely related condition frontotemporal dementia, or whether you think that this is targeting a disease mechanism which differentiates the two conditions?
Author Response
SOD1 has roles in glutamate excitotoxicity, and many other ALS genes such as TARDBP and FUS have roles in RNA processing. Could you include some discussion around these genes (and their encoded proteins) and their roles in the interaction of these two pathways? Could you also discuss any established or potential links (eg. are there any genomic features of these genes that flag them as being susceptible?) between ALS genes involved in RNA processing and A-to-I RNA editing?
We thank the reviewer for the valuable comment. We modified the sentences in section 3 (Page 8, line 298-325 in the revised manuscripts) as follows and added some references in order to facilitate readers’ understanding; reference numbers are 119, 120, 123, 124, 125, 126, 127, and 128.
“Page 8, line 298-325 in the revised manuscripts”
In addition to sporadic ALS, some ALS-linked genes are involved in excitotoxicity due to dysregulation of glutamatergic signaling or A-to-I RNA editing.
FUS is involved in RNA processing and it has been demonstrated that excitotoxicity affected FUS translocation from the nucleus to the cytoplasm and increased FUS-mediated dendritic GluA2 mRNA expression [119]. Additionally, glutamate release from brain synaptosome and glutamate uptake were increased in transgenic mice with mutant FUS lacking nuclear localization signal motif [120]. Regarding RNA editing, ADAR2 downregulation with a concomitant expression of Q/R site-unedited GluA2 mRNA was reported in the motor neurons of ALS patients carrying FUSP525L mutation [121], but not in motor neurons differentiated from iPSCs derived from ALS patients carrying FUSH517D mutation [113].
Furthermore, reduced ADAR2 activity due to the loss of nuclear ADAR2, rather than a decrease in ADAR2 expression, was reported the cause of a widespread reduction in RNA editing with ADAR2 cytoplasmic mislocalization in the motor neurons and differentiated motor neurons, which were generated from iPSCs derived from ALS patients carrying C9ORF72 with enhanced hexanucleotide repeat expansion [122]. Moreover, it has been demonstrated that dipeptide repeat proteins derived from hexanucleotide repeat expansion of C9ORF72 bind to ADAR1 and ADAR2, thereby inhibiting their RNA editing activity in vitro[123]. However, one study reported that all of GluA2 were edited at the Q/R site in motor neurons differentiated from iPSCs derived from ALS patients carrying C9ORF72 with enhanced hexanucleotide repeat expansion [124].
Although TDP-43 mislocalizes from the nucleus to the cytoplasm in the primary neuronal culture exposed to excitotoxicity [119]and may have a modulatory role in ADAR1-mediated A-to-I RNA editing in the cell lines [125], no studies have demonstrated aberrant RNA editing at the Q/R site in GluA2 in familial ALS carrying TARDBP mutations. In addition, possible roles of excitotoxicity in ALS-linked SOD-1 mutations have been suggested in SOD-1 transgenic mice [126,127], but all GluA2 mRNA expressed in the motor neurons of mutated human SOD-1 transgenic rats were edited at the Q/R site [128].
Can you comment on whether you think A-to-I RNA editing based therapeutics will be beneficial to all ALS patients or only a subset? If only a subset, how would you determine which patients would or would not benefit, and can this be effectively translated into a routine clinical test for patients?
We thank the reviewer for the valuable comment. We think A-to-I RNA editing-based therapeutic strategies are effective for both sporadic and some familial ALS that are related with dysregulation of RNA editing. Then, we added the sentences in section 4 (Page 9, line 356 in the revised manuscripts) and modified the sentences in section 4 (Page 11, line 404-407 in the revised manuscripts) as follows.
“Page 9, line 356”
not only sporadic ALS and some familial ALS
“Page 11, line 404-407 in the revised manuscripts”
The A-to-I RNA editing-based therapeutic strategies would effectively rescue the death of lower motor neurons in the majority of ALS patients, including patients with both sporadic and some familial ALS that are associated with dysregulation of A-to-I RNA editing.
Could you provide a comment around whether you think this sort of therapeutic approach would be applicable to the closely related condition frontotemporal dementia, or whether you think that this is targeting a disease mechanism which differentiates the two conditions?
We thank the reviewer for the valuable comment. As you have correctly pointed out, RNA editing-based therapy is limited to the death of lower motor neurons. Further studies are needed for elucidation whether the A-to-I RNA editing based therapeutic strategies are effective for death of upper motor neurons and more broadly, death of cortical neurons of frontotemporal lobar degeneration. Therefore, we added the sentences in section 4 (Page 11, line 407-410 in the revised manuscripts) as follows.
“Page 11, line 407-410 in the revised manuscripts”
However, as roles of dysregulation of RNA editing in the upper motor neuron death remain elusive, further studies are needed for elucidation of the pathogenic mechanism underlying death of upper motor neurons in ALS, and more broadly, death of cortical neurons in frontotemporal lobar degeneration, that is closely associated with ALS.

Reviewer 2 Report
The review entitled "RNA editing: a new therapeutic target in amyotrophic lateral 2 sclerosis and other neurological diseases" by Takashi Hosaka, Hiroshi Tsuji, and Shin Kwak is a timely review on a topic of great interest, especially when RNA is the focus of many diseases and created a new paradigm in treating human diseases. The review is well written and is highly informative. It has been presented in a very figurative manner to ease the understanding and readability of the content. Gene therapy, which is emerging as a new frontier as a new therapeutic opportunity for treating incurable diseases through RNA editing can also target neurological diseases which are also associated with RNA dysregulation which is also seen in ALS along with other neurodegenerative diseases. RNA editing in ALS, Ca2+ influx and excitotoxicity, and RNA dysregulation and therapy by RNA editing of dysregulation in ALS have been succinctly described, with thought-provoking ideas that can be utilized in the future. Although difficult to treat ALS, RNA editing holds better promise at the early stages of ALS than therapeutics that have not fared well in treating diseases.
Author Response
We thank the reviewer for the deep understanding and the valuable comments.
Reviewer 3 Report
The submitted review includes a summary of ADARs and their possible role in neurodegeneration and clinical pathology. There is also a chapter dedicated to ALS. In addition, accompanying figures summarize these data.
Overall, there is a need to revise manuscript content, with regard to the use of appropriate terms, e.g.:
L18 implicated=speculated?
L20 status epilepticus
L20 slow death=? Neurodegeneration?
Regarding manuscript content, 2/3 of the abstract could be the summary of an experimental work on site-specific A-I RNA editing of a metabotropic subunit transcript in ALS. It is not efficient or indicative of what follows. The authors did try to build the next chapters in a succinct manner.
However, an extensive recent review in the exact topic has been published in Methods in Molecular Biology [RNA Editing in Neurological and Neurodegenerative Disorders] by Cruz & Kawahara.
A different structural approach as in an older review in Frontiers in Genetics [Dysregulated A to I RNA editing and non-coding RNAs in neurodegeneration] by Minati Singh has been also used by Marceca et al to discuss recent advances in cancer, and by Destefanis in RNA Journal to discuss implications in a wide array of diseases [A mark of disease: how mRNA modifications shape genetic and acquired pathologies].
Practical implications have been recently discussed by Zhu et al in Gene Therapy for Neurodegenerative Disease: Clinical Potential and Directions, Frontiers in Molecular Neuroscience, 2021 and by Sun & Roy in Gene-based therapies for neurodegenerative diseases, Nature Neuroscience, 2021.
Despite the surplus of graphics, I believe that the submitted work does not provide complementary information. Therefore, it is not possible for us to provide advice as to improve this work or decide in favour of.
Author Response
Overall, there is a need to revise manuscript content, with regard to the use of appropriate terms, e.g.:
L18 implicated=speculated?
L20 status epilepticus
L20 slow death=? Neurodegeneration?
We think the reviewers for valuable comment. We changed these terms in line with reviewer’s points.
Regarding manuscript content, 2/3 of the abstract could be the summary of an experimental work on site-specific A-I RNA editing of a metabotropic subunit transcript in ALS. It is not efficient or indicative of what follows. The authors did try to build the next chapters in a succinct manner.
However, an extensive recent review in the exact topic has been published in Methods in Molecular Biology [RNA Editing in Neurological and Neurodegenerative Disorders] by Cruz & Kawahara.
A different structural approach as in an older review in Frontiers in Genetics [Dysregulated A to I RNA editing and non-coding RNAs in neurodegeneration] by Minati Singh has been also used by Marceca et al to discuss recent advances in cancer, and by Destefanis in RNA Journal to discuss implications in a wide array of diseases [A mark of disease: how mRNA modifications shape genetic and acquired pathologies].
Practical implications have been recently discussed by Zhu et al in Gene Therapy for Neurodegenerative Disease: Clinical Potential and Directions, Frontiers in Molecular Neuroscience, 2021 and by Sun & Roy in Gene-based therapies for neurodegenerative diseases, Nature Neuroscience, 2021.
Despite the surplus of graphics, I believe that the submitted work does not provide complementary information. Therefore, it is not possible for us to provide advice as to improve this work or decide in favour of.
We thank the reviewer for the valuable comments. We focused on the ALS-related excitotoxicity due to dysfunction of glutamatergic signaling resulting from dysregulated RNA editing at the GluA2 Q/R site, and derived promising future ALS therapy. Therefore, we added the sentences in the last paragraph in section 1 (Page 4, line 139-141 in the revised manuscripts) as follows and added some references in order to facilitate readers’ understanding; reference numbers are 41, 42, 43, and 44.
“Page 4, line 139-141 in the revised manuscripts”
as some recent excellent reviews have described about association of RNA editing with neurodegenerative diseases or current and promising gene therapies for neurodegeneration in general [41-44]
